# Sequential Inflammatory and Matrisome Programs Drive Remodeling of the Mouse Carotid–Jugular Arteriovenous Fistula

**DOI:** 10.3390/cells14241998

**Published:** 2025-12-16

**Authors:** Filipe F. Stoyell-Conti, Alexander M. Kaiser, Miguel G. Rojas, Yuntao Wei, Matthew S. Sussman, Juan S. Lopez-McCormick, Marwan Tabbara, Xiaofeng Yang, Roberto I. Vazquez-Padron, Laisel Martinez

**Affiliations:** 1DeWitt Daughtry Family Department of Surgery, Leonard M. Miller School of Medicine, University of Miami, Miami, FL 33136, USA; ffs9@med.miami.edu (F.F.S.-C.); alexander.kaiser@miami.edu (A.M.K.); mgr98@med.miami.edu (M.G.R.); ywei@med.miami.edu (Y.W.); sussmanm@miami.edu (M.S.S.); jsl303@med.miami.edu (J.S.L.-M.); mtabbara@med.miami.edu (M.T.); rvazquez@med.miami.edu (R.I.V.-P.); 2Department of Cardiovascular Sciences, Lewis Katz School of Medicine, Temple University, Philadelphia, PA 19140, USA; xiao-feng.yang@temple.edu; 3Research Service, Bruce W. Carter Veterans Affairs Medical Center, Miami, FL 33125, USA

**Keywords:** macrophage, fibroblast, myofibroblast, vein, arteriovenous fistula, inflammation, extracellular matrix

## Abstract

**Highlights:**

**What are the main findings?**
Macrophages and fibroblasts are the main populations responsible for postoperative remodeling of the vein in the mouse carotid–jugular arteriovenous fistula model.Postoperative macrophages activate venous fibroblasts via secretion of IL-1β, osteopontin, and TGF-β1. Fibroblasts, in turn, provide a rich microenvironment for macrophage recruitment, survival, and pro-inflammatory differentiation.

**What are the implications of the main findings?**
The mouse carotid–jugular arteriovenous fistula model reproduces the core inflammatory–fibrotic axis of fibroblast activation in humans, lending support to its application for mechanistic studies of AVF remodeling.

**Abstract:**

The mouse carotid–jugular arteriovenous fistula (AVF) is a widely adopted surgical model to study venous remodeling after AVF creation. Despite its increasing use, the extent to which this model recapitulates the cellular and molecular remodeling processes observed in humans remains uncertain, which is essential for validating its translational relevance. Using bulk and single-cell RNA sequencing, we have depicted the transcriptional and cellular evolution of the mouse jugular vein after AVF anastomosis. Global transcriptomic profiling revealed that venous remodeling begins with a robust inflammatory response, followed by a prominent extracellular matrix (ECM) remodeling phase that peaks at postoperative day 10. Single-cell analyses confirmed the role of macrophage (3-fold) and neutrophil infiltration (12-fold) in sustaining the onset of venous remodeling. These monocytes/macrophages exhibited marked upregulation of pro-inflammatory and pro-fibrotic genes, including *Il1b*, *Spp1*, *Fn1*, *Thbs1*, and *Tgfb1*. Evidence of the differentiation of fibroblasts into myofibroblasts positive for *Postn*, *Col8a1*, and *Thbs1* emerged by postoperative day 5. The temporal dynamics of differentially expressed genes in these myofibroblasts closely mirrored the ECM gene expression patterns identified by bulk RNA-seq, indicating that they are the principal source of ECM deposition in the AVF. Cell-to-cell communication analyses highlighted macrophages and fibroblasts as the main populations driving postoperative remodeling. Comparative analysis with single-cell data from human pre-access veins and AVFs demonstrated that the mouse model reproduces the core inflammatory–fibrotic axis of fibroblast activation observed in humans, supporting its utility for mechanistic studies of postoperative ECM remodeling.

## 1. Introduction

The mature arteriovenous fistula (AVF) remains the preferred vascular access for hemodialysis, offering superior long-term patency and fewer complications compared to alternative modalities [1]. Unfortunately, no therapeutic interventions currently exist to prevent the failure of approximately 40% of newly created accesses due to postoperative venous stenosis. The low success rate of clinical trials aimed at improving AVF maturation is now thought to stem, at least in part, from the misleading pre-clinical data derived from poorly characterized animal models.

Two mouse AVF models are commonly used in mechanistic studies that investigate the adaptive response of the vein to supra-arterial circulation: the carotid–jugular model, which involves an end-to-side anastomosis of the jugular vein to the carotid artery [2], and the aortocaval puncture model, in which the aorta is connected side-to-side to the inferior vena cava using a needle puncture [3]. These models enable the use of genetic knockout and knock-in strains for precise loss- and gain-of-function studies to test the involvement of specific genes in postoperative venous remodeling. Between them, the carotid–jugular AVF is the one that most closely resembles the human AVF in anatomic configuration. While these models and their surgical variations have provided important insights into the hemodynamic and molecular alterations after AVF creation [2,3,4,5,6,7,8,9], a comprehensive understanding of the cellular dynamics and gene expression programs driving AVF maturation remains incomplete.

In this study, we generated the first temporal transcriptional profiles and single-cell transcriptomic maps of the mouse carotid–jugular AVF. These data provide detailed insights into the roles of myofibroblasts, fibroblasts, and immune cells in the healing of the vascular wall and cellular differentiation after anastomosis. Notably, several key molecular changes identified in mice were also observed in human AVFs [10], underscoring the translational relevance of this model for investigating the fibrotic mechanisms driving AVF failure.

## 2. Materials and Methods

### 2.1. Mice and Surgical Procedures

Male and female C57BL/6 mice (The Jackson Laboratory, Bar Harbor, ME, USA) 20–23 g in weight were randomly allocated to bulk (*n* = 25) and single-cell RNA sequencing (scRNA-seq, *n* = 20). All mice underwent two consecutive surgeries: 5/6 nephrectomy [11] and AVF creation 3–4 weeks later by anastomosing the end of the jugular vein to the side of the carotid artery [2]. For bulk RNA-seq, animals were euthanized and AVFs were harvested at early (3.6 ± 2.4 days), mid (10.0 ± 0.7 days), and late (20.0 ± 1.8 days) time points after AVF creation. Blood flow was measured before tissue harvest using a Transonic probe (Transonic Systems, Ithaca, NY, USA; Appendix A). The early (5.5 ± 1.5 days) and late (26.1 ± 3.4 days) time points were also profiled by scRNA-seq. Contralateral jugular veins were sequenced as the control group. At the end of the protocol, animals presented a blood urea nitrogen of 52.8 ± 6.64 mg/dL, confirming the success of the 5/6 nephrectomy. All animal procedures were approved by the University of Miami Institutional Animal Care and Use Committee and adhered to the National Institutes of Health Guide for the Care and Use of Laboratory Animals.

### 2.2. Bulk RNA Sequencing

Tissues were ground to a fine powder in a Spex/Mill 6770 cryogenic grinder (SPEX SamplePrep, Metuchen, NJ, USA). RNA was extracted using Trizol and treated with DNase I as previously described [12]. Total RNA was quantified and qualified using the Agilent Bioanalyzer (Agilent Technologies, Santa Clara, CA, USA), ensuring an RNA integrity score > 5. Poly(A)-selected mRNA libraries were prepared by Azenta (Azenta US, Indianapolis, IN, USA) using an ultra-low input RNA sequencing protocol. Libraries were sequenced by Azenta on the Illumina HiSeq X (Illumina, San Diego, CA, USA) using a paired-end 150-bp configuration and an output of ~350 million reads per lane. Raw counts were normalized using the DESeq2 package in R Studio 2025.01, and genes with a base mean < 100 were filtered out from downstream differential gene expression analyses. Pathway and process enrichment analyses based on differentially expressed genes (DEGs; log_2_[fold change] ≥ 1, *p*_adj < 0.05) were performed using the Metascape tool [13]. Matrisome genes were identified from the curated and proteomics-validated blood vessel collection at the Matrisome Database [14]. Bioinformatic analyses of the bulk RNA-seq dataset were performed independently of the single-cell atlas and without data deconvolution.

### 2.3. Single-Cell RNA Sequencing

A pool of 10 AVFs or contralateral veins yielded 30,000–50,000 optimal cells for scRNA-seq. The cell suspensions were prepared by finely mincing the vessels and incubating them in 1X Vessel Dissociation Enzyme Solution (containing collagenase types XI and II, hyaluronidase, soybean trypsin inhibitor, DNase I, dispase, elastase, and HEPES) for 90 min at 37 °C as previously described [15]. The cell suspension was filtered through a 40-μm strainer, washed twice with Hank’s Balanced Salt Solution (HBSS), and resuspended in 0.1% BSA-PBS. Libraries were prepared using the Chromium Single Cell 3′ Library and Gel Bead Kit v3 (10X Genomics) targeting 10,000 cells per group and sequenced on the Illumina NovaSeq 6000 at the University of Miami John P. Hussman Institute for Human Genomics, Center for Genome Technology.

Data processing and analysis were performed using the Seurat package (Version 4.0.2) in R. Cells expressing fewer than 200 or more than 6000 genes, or with >10% mitochondrial gene expression, were excluded. Data were normalized using the “LogNormalize” method and using a scale factor of 10,000. Variable genes were identified using Seurat’s FindVariableGenes function. Principal component analysis was performed on variable genes, and the top 12 principal components were selected for further analysis. Uniform Manifold Approximation and Projection (UMAP) and clustering were performed with a resolution of 0.5. Marker genes for each cluster were identified using FindAllMarkers and the Wilcoxon rank-sum test. Pathway enrichment analysis was performed using the webpage-based tool Metascape [13]. Gene set analysis was conducted using GeneAgent [16] focusing on genes with an average log_2_[fold change] ≥ 1 and *p*_adj < 0.05. Functional scores were added to the integrated Seurat object using the function AddModuleScore in Seurat 4.0. Scores were based on the “GOBP_Leukocyte_Activation_Involved_In_Inflammatory_Response”, “GOBP_Positive_Regulation_Of_Extracellular_Matrix_Organization.v2025”, “COATES_Macrophage_M1_VS_M2_DN.v2025”, and “GAVISH_3CA_Metaprogram_Fibroblasts_Myofibroblasts.v2024” gene signature modules from the Molecular Signatures Database [17]. Ligand–receptor interactomes were analyzed using CellChat v2 [18]. Pseudotime trajectory analysis was performed as described in the Monocle3 vignette [19,20]. The top 100 transcription factors in fibroblasts were predicted based on the expression of target genes in the DoRothEA database [21].

We compared the gene expression changes occurring in murine mono/macros and myofibroblast/fibroblasts after anastomosis with those detected in the corresponding populations from human AVFs. We downloaded the single cell datasets of 6 human pre-access veins, 2 early AVFs (5–7 days postop), and 12 transposition AVFs (~90 days postop) from the Gene Expression Omnibus repository (accession number GSE305423). Human samples were processed and integrated independently of the mouse datasets, using conserved cell markers for cluster identification [10]. Differential gene expression analyses of mono/macros and myofibroblast/fibroblasts from veins and AVFs included the same number of cells randomly selected from the entire mono/macro and myofibroblast/fibroblast clusters of each experimental group and species. Common DEGs across species included genes expressed in >30% of the cells with log_2_[fold change] ≥ 1 and *p*_adj < 0.01.

## 3. Results

### 3.1. The Mouse AVF Remodels Through Waves of Inflammation and ECM Deposition

We first performed bulk RNA sequencing of the native contralateral vein and the outflow vein of the AVF collected at early (3.6 ± 2.4 days), mid (10.0 ± 0.7 days), and late (20.0 ± 1.8 days) time points after anastomosis to identify the global transcriptional changes during remodeling of the mouse AVF. The vein, prior to AVF creation, was predominantly characterized by transcriptional programs associated with homeostatic vascular functions, including smooth muscle cell (SMC) contraction, circulatory system regulation, and actin filament-based processes, compared with the postoperative groups (Figure 1A). Exposure of the vein to supraphysiological conditions significantly altered the expression of a fourth of the transcriptome (4541 of 20,094 captured genes) (Figure 1A). Differentially expressed genes (log_2_[fold change] ≥ 1, adjusted *p* < 0.01) revealed two distinct transcriptional states after anastomosis: an early postoperative state characterized by strong inflammatory activation, and a mid-to-late state dominated by wall remodeling and healing. The early postoperative state showed upregulation of myeloid or lymphoid markers (e.g., *Lyz2*, *Csf3r*, *Cx3cr1*), complement genes (e.g., *C1qa*, *C1qb*, *C1qc*), and cytokines (e.g., *Spp1*, *Il1b*, *Cxcl5*) (Figure 1A,B, Appendix A). This early response was also accompanied by a marked decline in canonical endothelial (e.g., *Pecam1*, *Vwf*) and SMC markers (e.g., *Myh11*, *Cnn1*, *Lmod1*) that persisted during remodeling.

Inhibitors of TGF-β signaling and pro-collagen processing (e.g., *Ltbp4*, *Pcolce2*, *Dcn*) [22,23,24], as well as metalloproteinase activators (*Mmp3*) [25], decreased acutely after anastomosis to set the stage for new ECM deposition (Figure 1B). By postoperative day 3, there was a balance of metalloproteases and their inhibitors (*Mmp8*, *Adam12*, *Timp1*), along with the upregulation of pro-fibrotic factors (*Spp1*, *Tgfb1*), ECM cross-linking enzymes (*Loxl2*), and genes involved in the production of a provisional ECM (*Has2*, *Vcan*) [26,27]. By day 10, gene activity associated with vascular wall healing/remodeling became even more evident. Expression of periostin (*Postn*) peaked, indicating maximal myofibroblast activation [28,29]. Accordingly, there was significant upregulation of fibrillar collagens (*Col1a1*, *Col3a1*), elastin (*Eln*), fibronectin (*Fn1*), and additional crosslinking enzymes (*Lox*, *Loxl1*) that remained high in late fistulas. The expression of basement collagens (*Col8a1*, *Col4a1*) and thrombospondins (*Thbs1*, *Thbs4*) peaked at 21 days, likely reflecting the stabilization of cell-ECM interactions and enhanced mechanotransduction. Upregulation of the mechanosensitive machinery also reflects the ability of the vein to continue remodeling. In fact, additional matrisome genes involved in exacerbated ECM deposition (*Prelp*, *Comp*) and inhibition of metalloproteinases (*Timp3*) were significantly upregulated during this late remodeling period (Figure 1B,C).

### 3.2. Immune Cell Infiltration and Activation of Fibroblasts Explain the Stages of Venous Remodeling

To better define the cell populations contributing to the early and late remodeling of the mouse AVF, we performed scRNA-seq of contralateral jugular veins and the outflow veins of early (5.5 ± 1.5 days postop) and late AVFs (26.1 ± 3.4 days) (Figure 2A–C). Similarly to human veins and AVFs [10], the most abundant populations in the three experimental groups were fibroblasts (range 20.7–44.1%), monocyte/macrophages (15.7–49.2%), and T cells (3.1–12.1%). We identified a de novo infiltration of mono/macros (3-fold increase) and neutrophils (12-fold) in early AVFs that explains the inflammatory burst inferred by bulk RNA-seq (Figure 2C). The proportion of fibroblasts increased modestly from 38.2% in contralateral veins to 44.1% in AVFs, while SMCs increased from 3.1 to 7.1%. Postoperatively, the latter were characterized by significant upregulation of stress-response genes (e.g., *Spp1*, *S100a4*, *Atf3*), likely due to mechanical stretching and high oxygen conditions (Appendix A). In contrast, endothelial cells (ECs) decreased in proportion after AVF creation (14.5 to 1.5%). In those that remained, a lower expression of canonical EC markers (e.g., *Vwf*, *Flt1*) and upregulation of contractile genes (e.g., *Acta2*, *Tagln*) suggested an endothelial-to-mesenchymal transformation (Appendix A).

The global changes in cellular functions in the murine jugular vein were characterized using gene signature scores (Figure 2D,E). The inflammatory leukocyte activation score was elevated in mono/macros of early AVFs and remained high in late AVFs compared with the contralateral veins. Neutrophils were also highly inflammatory but present in lower proportions. Concurrently, the score of ECM organization illustrated the acute activation of fibroblasts after anastomosis and a progressive increment from contralateral veins to late AVFs. There were no significant changes in the DNA synthesis and cell division scores after anastomosis (Appendix A). These findings indicate that the postoperative upregulation of matrisome genes observed by bulk RNA-seq is due to the phenotypic activation of ECM-producing fibroblasts and mural cells more than cell proliferation. Importantly, the temporal dynamics of inflammation and ECM remodeling scores in the mouse AVF mimic the postoperative profiles of single cell populations from early (5–7 days postop) and late (~90 days) human fistulas [10].

### 3.3. Macrophages Undergo a Pro-Fibrotic Reprogramming After AVF Creation

Next, we re-clustered mono/macros at resolution 0.5 to identify two transcriptionally distinct phenotypes orchestrating the remodeling of the murine AVF (Figure 3A–D). Mono/Macro-1 cells were the predominant type in contralateral veins and expressed a gene program consistent with immune regulation and resolution of inflammation. This subset was defined by upregulation of chemokines (e.g., *Ccl4*, *Ccl8*, *Cxcl10*), heat shock proteins (e.g., *Hspd1*, *Hspb1*, *Dnajb1*), complement genes (e.g., *C1qa*, *C1qb*, *C1qc*), interferon-stimulated genes (e.g., *Ifit1*, *Ifit2*, *Oasl1*), and oxidative stress regulators (e.g., *Hpgds, Txnip*). The high expression of the transcription factors *Atf3* and *Cited2* further supports a stress-adaptive, homeostatic phenotype (Figure 3C) [30,31,32]. Mono/Macro-1 cells decreased significantly after anastomosis but seemed to participate in the resolution of inflammation in late AVFs, as illustrated by the elevated M2-like polarization score and the upregulation of *Mrc1*, *Klf4*, *Irf4*, *Cd36*, and *Retnla* compared with early fistulas (Figure 3E,F and Appendix A). The gene repertoire of this late pro-resolving phenotype also has unique antioxidant adaptations compared with the Mono/Macro-1 of contralateral veins, including the higher expression of *Hmox1*, *Sqstm1*, and *Hspa1a* (Appendix A). Mono/Macro-2 cells, in turn, increased significantly after anastomosis and displayed a highly pro-inflammatory and tissue-remodeling program (Figure 3D,E). This included upregulation of TNF and IL-1 signaling mediators (e.g., *Traf1*, *Tnfrsf1b*, *Il1b*), cytokine receptors (e.g., *Csf2rb*, *Il7r*), and pattern-recognition receptors (e.g., *Clec4d*, *Clec4e*). Elevated expression of *Mmp12* and *Mmp19* reflect an ECM remodeling function; while *Acod1*, *Txnrd1*, and *Prdx1* indicate metabolic reprogramming and redox regulation. Additional upregulated genes, such as *Plaur*, *Fn1*, *Ptgs2*, *Spp1*, and *Thbs1*, highlight the role of this phenotype in adhesion, fibrosis, and inflammation (Figure 3D) [32,33,34,35].

Due to both changes in phenotype proportions and in transcriptional programs of individual cells over time (Figure 3B,E,F), over 200 DEGs were detected in mono/macros from early (280) and late AVFs (219) compared with those from the contralateral veins (Figure 4A,B, Appendix A). Eighty-eight of the upregulated DEGs and 70 of the downregulated ones were shared in common between both time points and were mostly associated with a postoperative enhancement of inflammatory processes (e.g., up in AVFs: *Spp1*, *Il1b*, *Lgals3*, *Ptgs2*) and a reduction in homeostatic functions (e.g., down in AVFs: *Cd163*, *Lyve1*, *F13a1*, *Gas6*) (Figure 4C–E). The pathway analysis of the late time point demonstrated an enrichment in some regulatory functions associated with resolution of inflammation (Figure 4D).

We then compared the DEGs in early and late mono/macros (vs. collateral veins) with those detected in the corresponding populations from early (5–7 days postop) and late (~90 days) human fistulas (Appendix A). We focused on genes expressed in >30% of the cells with log_2_[fold change] ≥1 and *p*_adj < 0.01. Twenty-eight percent of upregulated DEGs in murine mono/macros at the early time point were also upregulated in early human AVFs compared with pre-access veins. These consisted, for the most part, of inflammatory mediators associated with tissue remodeling (e.g., *Il1b*, *Spp1*, *Thbs1*, *Ptgs2*). Similarly, 13% of upregulated genes in late mono/macros of mice were upregulated in late human fistulas. Many of the upregulated inflammatory and pro-fibrotic mediators from early AVFs (e.g., *Il1b*, *Fn1*) were included in this set, as well as genes associated with resolution of inflammation (e.g., *Il10*, *Il1rn*, *Trem2*, *Apoe*, *Sod2*, *Prdx1*, *Fth1*, *Metrnl*). Among the downregulated DEGs in mouse mono/macros, 20% at the early time point and 11% in late AVFs were downregulated in the corresponding groups of human fistulas (Appendix A). These included several homeostatic genes (e.g., *Lyve1*, *F13a1*, *Gas6*), the transcription factor *Maf*, which determines the anti-inflammatory polarization of mono/macros [36,37,38], and scavenger receptors (e.g., *Mrc1*, *Cd163*). These comparisons identified conserved molecular changes in the immunoregulation of venous remodeling in response to surgical trauma and/or supraphysiological flow.

### 3.4. Fibroblasts Transform from a Quiescent to an Activated State After AVF Creation

Fibroblasts are more abundant in the vein than mural cells and play a critical role in postoperative hemostasis, wall healing, and reorganization of the ECM. Therefore, we studied the temporal differentiation of fibroblasts in mouse AVFs and their similarities to those from human fistulas. After re-clustering at resolution 0.5, three transcriptionally distinct fibroblast subsets (FB-1 to 3) were identified that changed in abundance and function during AVF remodeling. In contralateral veins, fibroblasts were largely quiescent, with FB-1 representing more than 80% of the population (Figure 5A,B). FB-1 cells were characterized by genes that maintain ECM integrity (e.g., *Eln*, *Dcn*, *Podn*), vascular stability (e.g., *Gas6*, *Abcc9*, *F3*), and redox balance (e.g., *Apoe*, *Txnip*, *Inmt*) (Figure 5C, Appendix A). In early AVFs, FB-2 cells expanded nearly ten-fold, representing ~50% of the overall fibroblast cluster (Figure 5B). Gene signature scores and trajectory analyses identified this subpopulation as myofibroblasts originating from FB-1 cells and from a minor fibroblast subtype, named FB-3 (Figure 5D,E). During this differentiation, FB-2 cells gained significant expression of contractile markers (e.g., *Tagln*, *Acta2*, *Tpm2*), mechanosensitive genes (e.g., *Thbs1*, *Thbs4*), and ECM proteins (e.g., *Postn*, *Col8a1*, *Col1a1*, *Tnc*). Upregulated protein chaperones such as *P4hb, Serpinh1*, and *Pdia6* facilitate collagen maturation, while metalloproteinases and inhibitors (e.g., *Mmp14*, *Timp1*) regulate ECM turnover (Figure 5C, Appendix A). The differentiation of FB-2 cells continued in late AVFs, with further upregulation of basement collagens, mechanosensitive genes, complement factors, cytokines, and gelsolin (*Gsn*) to support cell migration and immune activation (Appendix A) [39,40]. In contrast to the early postoperative expansion of FB-2 cells, FB-3 cells remained in low proportion in early AVFs but increased in late fistulas. Transcriptionally, they were characterized by upregulation of genes involved in immune regulation and metabolic signaling (Figure 5C, Appendix A). These include regulators of inflammation and complement activity (e.g., *Cd55*, *Tnfaip6*), and enzymes involved in glycosaminoglycan biosynthesis (e.g., *Ugdh*, *Uap1*, *Gfpt2*). The relative abundances of fibroblast subtypes over time and their temporal profiles of matrisome gene expression (Appendix A) demonstrate that, in terms of magnitude, FB-2 cells are the main population responsible for postoperative ECM deposition.

Due to the changes in proportions and in the expression profiles of individual cells, a total of 226 DEGs were detected in fibroblasts from early AVFs and 138 in late fistulas compared with those from contralateral veins (Figure 6A, Appendix A). Fifty-one of the upregulated DEGs were found at both time points and included markers of myofibroblast differentiation (e.g., *Postn*, *Acta2*, *Runx1*), mechanosensitive genes (e.g., *Thbs1*, *Vcam1*), and inflammatory transcriptional regulators (e.g., *Nfkb1*, *Atf3*) (Figure 6B,C). Thirty-six of the downregulated DEGs were also shared by both early and late fibroblasts. These included regulators of actin assembly (e.g., *Gsn*, *Tmsb10*), complements and coagulation factors (e.g., *C7*, *F3*), and inhibitors of TGF-β signaling (e.g., *Fmo2*, *Gdf10*). The pathway analyses illustrated the shifts in fibroblast functions from the native vein to the late AVF, which concur with the matrisome changes reported by bulk RNA-seq (Figure 1 and Figure 6D). Fibroblasts of contralateral veins showed activation of hemostasis and the complement cascade, but negative regulation of cell migration. In contrast, processes associated with ECM remodeling, cell motility, and responses to wounding were enriched in early fibroblasts. Late fibroblasts showed enrichment of mechanosensitive pathways and inflammatory activation. Considering that the peak of vascular inflammation occurs during the early time point, the pathways enriched in late fibroblasts suggest a chronic inflammatory activation and/or phenotypic differentiation. An inflammatory basis for either of these processes is supported by the analysis of the top 100 transcription factors activated in early and late fibroblasts based on the expression of target genes (Appendix A). This analysis predicted maximum activation of the TGF-β signaling factors Smad3 and 4 in myofibroblasts (FB-2) of early AVFs and a progressive increase in Nfkb1 and Rela activity in FB-2 and FB-3 cells over time.

Comparison of DEGs in early and late murine fibroblasts (vs. collateral veins) with DEGs in the corresponding populations from early and late human fistulas uncovered significant similarities, particularly in the genes expressed by the myofibroblast (FB-2) subpopulation (Appendix A, Appendix A). These similarities were observed in both the upregulated and downregulated directions. Thirty-two percent of upregulated DEGs and 37% of downregulated genes in the overall fibroblast cluster of mice were differentially expressed in the same direction in early human AVFs compared with pre-access veins. Similarly, 21% of upregulated genes and 17% of downregulated DEGs in late fibroblasts of mice were shared in common with late human fistulas. Altogether, these findings demonstrate that, in mice and humans, the differentiation of fibroblasts into myofibroblasts is a dominant process during postoperative AVF remodeling.

### 3.5. Cell-to-Cell Communication Analyses Single out Macrophages and Fibroblasts as the Drivers of Postoperative Remodeling

We performed global cell-to-cell communication analyses to infer the strength and direction of signaling interactions contributing to the postoperative remodeling of the mouse AVF. The total number and strength of ligand–receptor interactions increased progressively after anastomosis to reach the highest levels in late AVFs (Figure 7A and Appendix A). Postoperatively, fibroblasts and mono/macros were the top senders of signals (ligand secretion), while mono/macros and neutrophils were the top receivers (receptor binding) (Figure 7A). The top ligands secreted by fibroblasts of early AVFs compared with those from contralateral veins were periostin, pleiotrophin (PTN), angiopoietin-like proteins (ANGPTLs), midkine (MK), macrophage migration inhibitory factor (MIF), and chemokines (Figure 7B). These outgoing signals remained enriched in fibroblasts from late fistulas and were the top incoming interactions for macrophages of both postoperative time points (Appendix A). Concurrently, osteopontin (SPP1) originating from mono/macros was the top incoming signal in both early and late AVFs (Figure 7B–D). Other significant incoming interactions compared with contralateral veins included TGF-β and IL-1 (Figure 7B–D and Appendix A). The former was mostly derived from Mono/Macro-2 cells and myofibroblasts (FB-2), while Mono/Macro-2 and neutrophils were the main sources of IL-1 (Figure 7D). Altogether, these analyses uncovered a unique macrophage–fibroblast crosstalk in early and late AVFs that not only provides for the survival and differentiation of macrophages but also perpetuates the activation of fibroblasts.

## 4. Discussion

The murine carotid–jugular AVF model has become increasingly prevalent in mechanistic studies aimed at dissecting the processes by which the vein successfully adapts to supraphysiological conditions after AVF creation. Leveraging novel omics technologies, this study provides a comprehensive transcriptional characterization of venous remodeling in this model in nephrectomized mice, critically evaluating the extent to which it recapitulates clinically relevant molecular pathways of AVF maturation. We have validated the value of the carotid–jugular AVF model to study (1) the early surge of inflammation mediated by monocytes/macrophages and neutrophils, (2) the chronic inflammatory activation of fibroblasts and their differentiation into ECM-producing myofibroblasts, and (3) the establishment of a multifaceted macrophage–fibroblast interaction network that both initiates and regulates ECM deposition.

According to our transcriptomic study, the mouse carotid–jugular AVF represents an excellent model to dissect the mechanisms by which early inflammation after anastomosis drives venous remodeling. As we recently described in humans [10], neutrophils and mono/macros predominate during the early postoperative phase in the murine AVF, together accounting for nearly 70% of all captured cells in the vessel wall. The abundance of these cells reflects the de novo infiltration of myeloid cells into the injured vasculature to orchestrate the debridement and reconstruction of the vein after hemodynamic and surgical trauma. Accordingly, depletion of macrophages using clodronate-containing liposomes impaired the adaptive thickening of the vascular wall [41], a process that is essential for vascular resilience.

While most prior studies exploring macrophage involvement in venous remodeling have relied on the reductionist M1/M2 polarization model, which is now considered oversimplified, our data support the existence of distinct inflammatory and reparative macrophage populations in the murine AVF. The reparative subset, designated Mono/Macro-1, expressed stress-adaptive and immunoregulatory genes, including complement components and interferon-stimulated transcripts. These cells were the most abundant in the contralateral veins, suggesting a role in maintaining vascular homeostasis. On the other hand, the pro-inflammatory macrophage subset Mono/Macro-2 that remained elevated in the murine AVF until the end of the experimental time exhibited a highly inflammatory and pro-fibrotic transcriptional profile characterized by elevated expression of *Il1b*, *Fn1*, *Spp1*, *Thbs1*, *Thbs4*, and *Ptgs2*. Notably, the upregulation of mechanosensitive thrombospondin genes in *Spp1*^+^ Mono/Macro-2 suggests a mechanistic trigger that integrates ECM-derived mechanical cues with inflammatory signaling, enhancing immune cell recruitment, promoting IL-1β activation, and facilitating TGF-β1 release [42,43]. Higher proportion of pro-inflammatory mono/macros was associated with AVF failure in human AVFs [10], underscoring their potential contribution to venous stenosis.

Our study also demonstrates the utility of the carotid–jugular AVF model to investigate the role of fibroblasts in venous remodeling. We observed striking phenotypic transitions of venous fibroblasts after anastomosis. The homeostatic FB-1 fibroblasts, which were abundant in the pre-operative mouse vein, likely transitioned into *Postn*^+^ myofibroblasts (also known as FB-2) in response to biomechanical stress and inflammatory cues. This postoperative fibroblast population expands markedly and adopts a fibrotic and inflammatory phenotype characterized by the upregulation of collagens, matricellular proteins, and chemokines during postoperative venous remodeling. Like postoperative macrophages, *Postn*^+^ myofibroblasts constitute an important source of TGF-β within the fistula wall. Notably, one of the defining markers of *Postn*^+^ myofibroblasts is *Col8a1*, whose upregulation has been associated with maturation failure of human AVFs [12]. Importantly, our analysis does not exclude a potential contribution of SMCs to this population, although a definitive validation will require single-cell lineage tracing coupled with histological analyses.

Several macrophage–fibroblast interactions upregulated in the present study were enriched in association with human AVF failure [10]. These include osteopontin (SPP1), periostin (POSTN), TGF-β, angiopoietin-like factors (ANGPTL), and IL-1. These common pathways across species suggest that they correspond to a physiological remodeling process that simply goes overboard in failed human fistulas. In addition to their association with AVF failure [10], Spp1^+^ macrophages are a source of pro-fibrotic factors in the heart [44]. Various ANGPTLs also play a fibrogenic role in the heart and kidneys by enhancing TGF-β signaling and/or binding to integrins [45,46,47]. The specific role of these pathways in AVF remodeling and fibrosis should be further validated using genetic and pharmacologic approaches, while taking into consideration the molecular characteristics of clinical samples that are not replicated in mice. While the core inflammatory–profibrotic axes are upregulated in both species after anastomosis, there are still significant differences at the transcriptional level across species that may signify positive or negative regulatory loops. The breadth of molecular signals in humans reflects the demographic and clinical variability of the population, the different maturation outcomes after AVF surgeries, and the effects of decades of chronic kidney disease. As more omics data of human samples become available, we may be able to discern the molecular signatures of AVF failure in specific patient populations, which would improve the precision of mechanistic models and the translational value of the results.

The murine AVF model exhibits important hemodynamic and vascular differences compared with human accesses, including thinner vascular walls, the absence of venous valves, the central anatomical placement of the fistula, and a higher incidence of thrombosis, often unrelated to stenosis, stemming from traumatic vascular injury and/or species-specific coagulation characteristics. Moreover, AVF maturation failure is a clinical concept defined by access usability for hemodialysis and cannot be extrapolated to this experimental setting. Instead, the mouse model remains valuable for studying postoperative venous remodeling using surrogate parameters such as wall thickness, luminal area, and blood flow. Keeping this in mind, the availability of knock-out, knock-in, and humanized mouse strains enables the rigorous molecular dissection of pathways involved in vascular remodeling which, when integrated with human datasets, can effectively guide the design of mechanistic and pre-clinical studies in large animal models.

While our study provides valuable insights about the cellular and molecular mechanisms driving AVF remodeling in mice, several limitations must be acknowledged. First, as mentioned above, the mouse model lacks a standardized definition of stenosis, limiting its ability to accurately simulate human maturation failure. Second, there are profound differences between human and murine AVFs in the cellular heterogeneity of fibroblasts, mural cells, ECs, and mono/macro populations, reflecting the greater histological complexity of human veins. Third, the number of human veins and AVFs from which transcriptional data is available for cross-species comparisons is relatively small and not representative of the entire patient population. Fourth, the macrophage–fibroblast pathways implicated in postoperative inflammation and fibrosis by our analyses require functional validation. Despite these limitations, our study demonstrates that, as in humans, remodeling of the carotid–jugular AVF in mice is orchestrated by early immune infiltration, fibroblast activation, and loss or reprogramming of ECs. Comparisons with human data indicate that the murine model recapitulates the core inflammatory–fibrotic axis but lacks the overall complexity of human remodeling. Therefore, integrating insights from both biological systems will be instrumental for designing mechanistic studies of ECM remodeling and inflammatory signaling within a clinically relevant framework, minimizing the risk of misleading interpretations.

## 5. Conclusions

The mouse carotid–jugular AVF is a widely adopted surgical model that is used to study venous remodeling after AVF creation. While most animal studies in the past have focused on the development of venous intimal hyperplasia, published human evidence points to postoperative fibrosis as the main mechanism underlying AVF stenosis. Using temporal scRNA-seq analysis, we demonstrate that the mouse carotid–jugular AVF model recapitulates the core inflammatory–fibrotic axis of fibroblast activation observed in humans, lending support to its use for mechanistic studies of postoperative ECM deposition.

## Figures and Tables

**Figure 1 cells-14-01998-f001:**
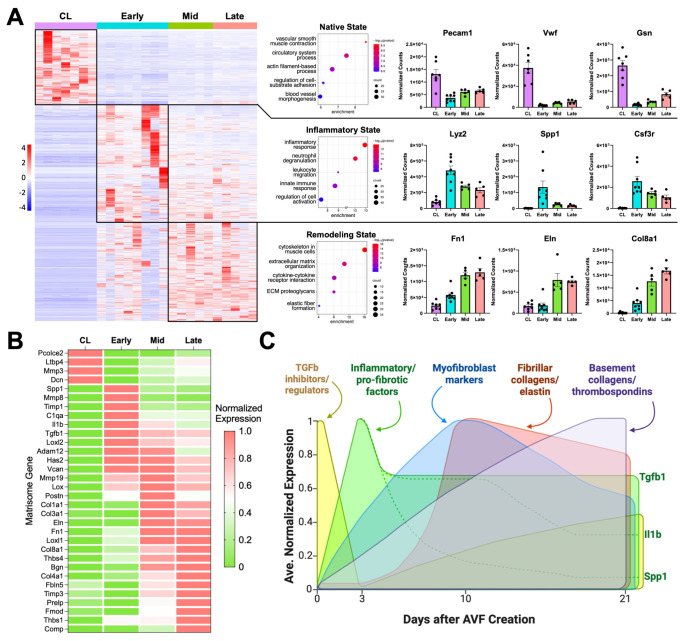
Transcriptional regulation of postoperative venous remodeling in the mouse carotid–jugular arteriovenous fistula (AVF) per bulk RNA sequencing analyses: (**A**) Heatmap of 4541 differentially expressed genes among contralateral jugular veins (CL, *n* = 7), and outflow veins from early (~3 days postop, *n* = 8), mid (~10 days, *n* = 5), and late AVFs (~21 days, *n* = 5). Pathway enrichment analyses and representative upregulated genes of the three transcriptional states identified in the heatmap. (**B**) Heatmap of selected differentially expressed matrisome genes illustrating the transcriptional stages of extracellular matrix (ECM) remodeling in the AVF. The average gene expression per experimental group was normalized so that the lowest level across the groups was set at 0 and the highest level at 1. (**C**) Temporal diagram of the main stages of ECM remodeling in the AVF based on the differential expression of matrisome genes.

**Figure 2 cells-14-01998-f002:**
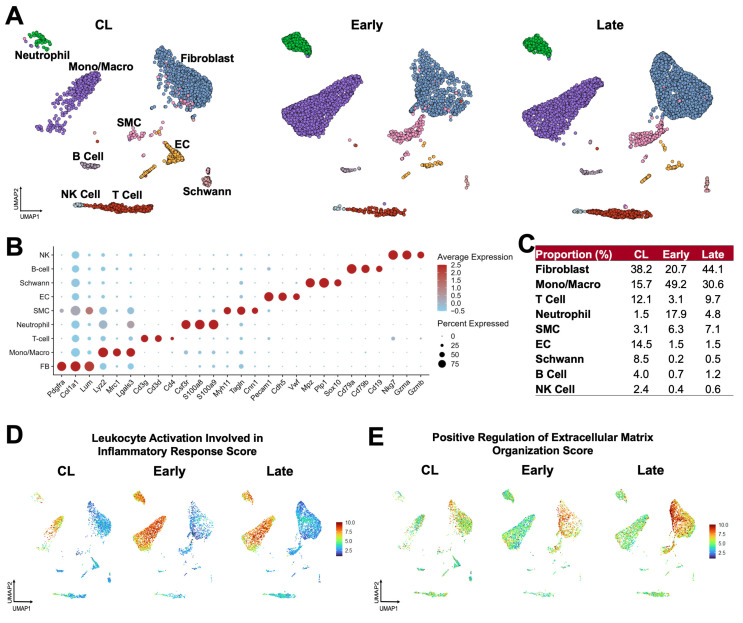
Postoperative venous remodeling in the mouse carotid–jugular arteriovenous fistula (AVF) per single-cell RNA sequencing: (**A**) Uniform manifold approximation and projection (UMAP) plot of 10,500 cells isolated from contralateral veins (CL, *n* = 10) and the outflow veins from early (~5 days postop, *n* = 10) and late AVFs (~26 days postop, *n* = 10). (**B**) Markers used for cell cluster identification. The color of the dots indicates the average gene expression per cluster with red signifying high expression, while the size of the dots represents the percentage of cells in the cluster expressing the gene. (**C**) Relative proportions of cell clusters per experimental group. (**D**,**E**) Functional profiling of single cell clusters in the experimental groups according to gene signature scores. Scores combine the expression levels of multiple genes with an experimentally curated function.

**Figure 3 cells-14-01998-f003:**
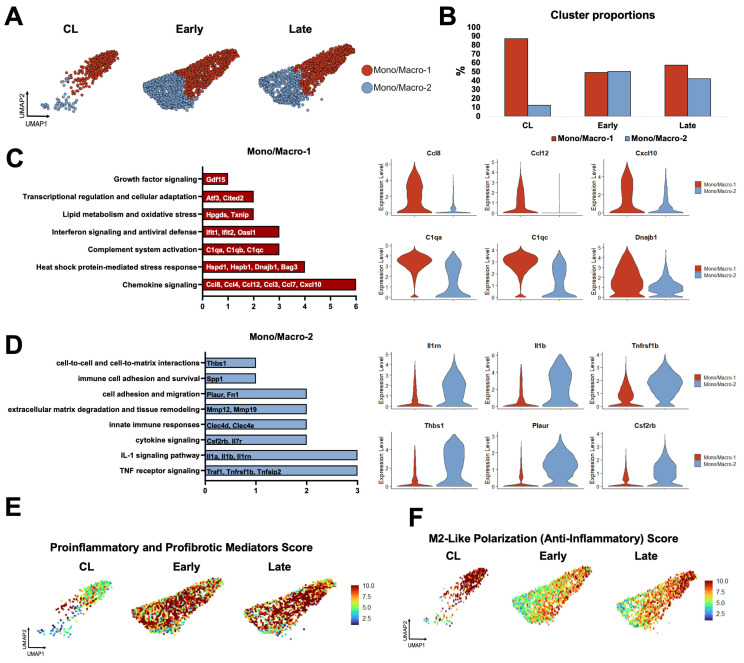
Monocyte/macrophage phenotypes during remodeling of the mouse arteriovenous fistula (AVF): (**A**,**B**) Focused UMAPs and relative proportions of monocyte/macrophage phenotypes in contralateral veins (CL) and the outflow veins from early and late AVFs. (**C**,**D**) Pathway enrichment analyses and representative differentially expressed genes between the two mono/macro subpopulations. (**E**,**F**) Functional profiling of mono/macros over time according to gene signature scores. Scores combine the expression levels of multiple genes with an experimentally curated function.

**Figure 4 cells-14-01998-f004:**
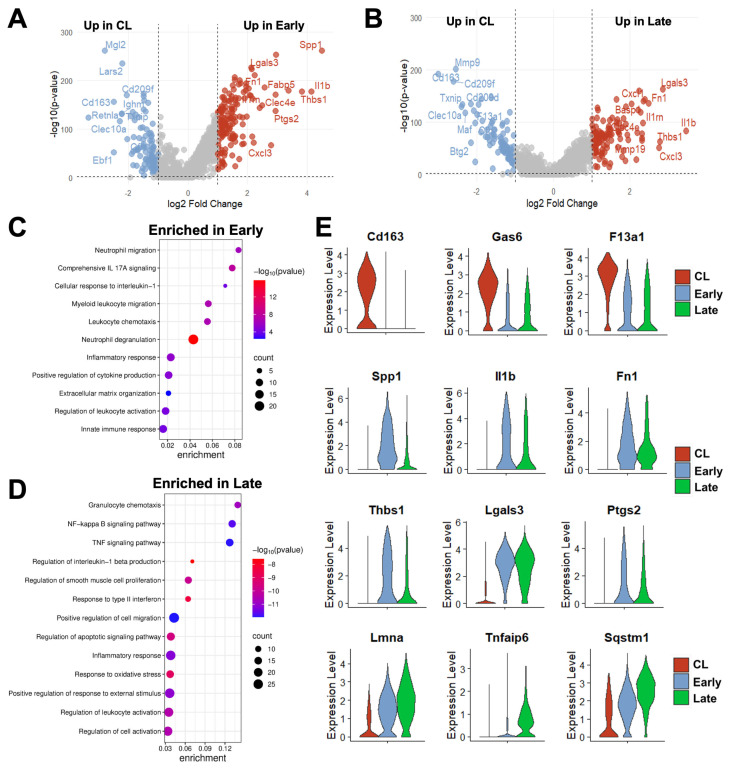
Time-dependent transcriptional changes in murine monocyte/macrophages after arteriovenous fistula (AVF) creation: (**A**,**B**) Volcano plots of differentially expressed genes between mono/macros from early or late AVFs and those from contralateral veins (CL). (**C**,**D**) Pathway enrichment analyses of mono/macros from early and late AVFs. (**E**) Time-dependent changes in the RNA expression of selected genes in mono/macros after AVF creation.

**Figure 5 cells-14-01998-f005:**
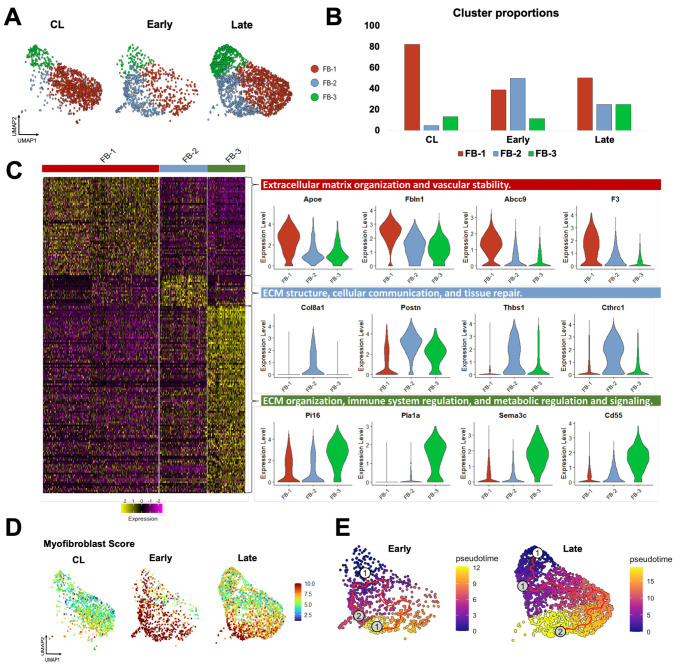
Fibroblast phenotypes during remodeling of the mouse arteriovenous fistula (AVF): (**A**,**B**) Focused UMAPs and relative proportions of fibroblast phenotypes in contralateral veins (CL) and the outflow veins from early and late AVFs. (**C**) Heatmap and representative examples of differentially expressed genes among the three fibroblast phenotypes. (**D**) Fibroblast activation and/or differentiation after AVF creation according to the myofibroblast gene signature score. (**E**) Pseudotime trajectory analyses of fibroblasts from early and late AVFs, ordering cells along the most likely path of myofibroblast differentiation. The white dot indicates the root of the trajectory, while the gray dots are the main leaves.

**Figure 6 cells-14-01998-f006:**
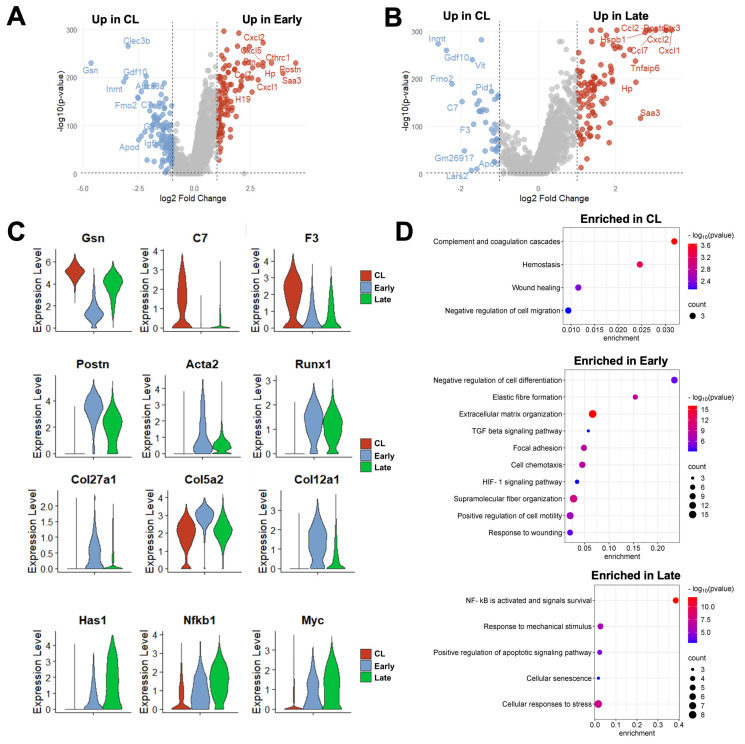
Time-dependent transcriptional changes in murine fibroblasts after arteriovenous fistula (AVF) creation: (**A**,**B**) Volcano plots of differentially expressed genes between fibroblasts from early or late AVFs and those from contralateral veins (CL). (**C**) Time-dependent changes in RNA expression of selected genes in fibroblasts after AVF creation. (**D**) Pathway enrichment analyses of fibroblasts from the three experimental groups.

**Figure 7 cells-14-01998-f007:**
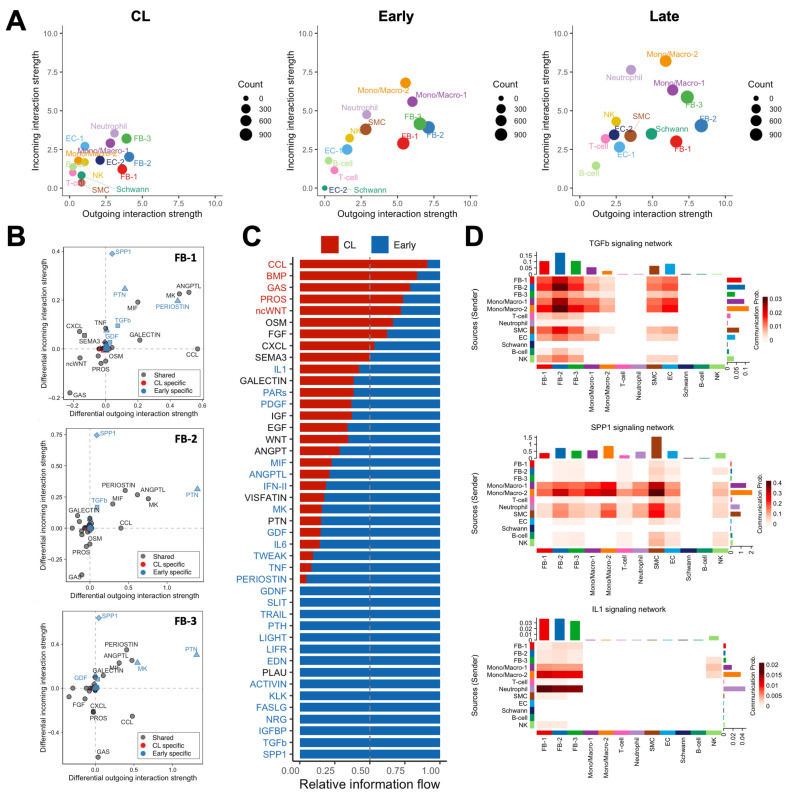
Macrophage–fibroblast crosstalk during remodeling of the mouse arteriovenous fistula (AVF): (**A**) Global cell-to-cell communication analyses in contralateral veins (CL) and outflow veins from early and late AVFs. The x- and y-axes indicate the strengths of outgoing (ligand secretion) and incoming interactions (receptor binding), respectively, while the sizes of the circles represent the number of predicted interactions. (**B**) Differential analysis of interactions in the three fibroblasts phenotypes from early AVFs compared with those from contralateral veins. Positive values in the x- and y-axes indicate outgoing and incoming interactions, respectively, that are significantly enhanced in early AVFs. Ligands in blue denote pathways that are specific in early AVFs; those in red are specific in contralateral veins; and ligands in gray indicate pathways that occur in both. Triangles indicate outgoing specific; squares, incoming specific; diamonds, incoming- and outgoing-specific; circles, occur in both. (**C**) Ranking of cell–cell interactions originating from all cell populations in early AVFs and received by fibroblasts. The red and blue fonts in the ligand names indicate interactions that are statistically significant in contralateral veins and early AVFs, respectively. (**D**) Heatmap of TGF-b, SPP1, and IL-1 mediated interactions in early AVFs. Senders of ligands are shown on the left, while the receivers are shown at the bottom of each heatmap.

## Data Availability

Sequencing data were deposited in the Gene Expression Omnibus.

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
