# Peer review of "Sequential Inflammatory and Matrisome Programs Drive Remodeling of the Mouse Carotid–Jugular Arteriovenous Fistula"

_cells, 2025, doi:10.3390/cells14241998_

Round 1
Reviewer 1 Report
Comments and Suggestions for Authors
The manuscript employs an exceptionally thorough and state-of-the-art research approach to modeling AVF and notably integrates both bulk and single cell RNA sequencing data. The strengths of the study lie in its experimental precision, detailed temporal sampling, and comparative human data analysis, which clearly enhance the translational value of the work. The in-depth analysis of macrophage–fibroblast interactions is particularly noteworthy, revealing important new mechanistic insights into AVF remodeling.
A few suggestions that could further strengthen the manuscript:
It would be valuable to discuss in more detail the functional validation of the identified macrophage phenotypes and which components of macrophage-fibroblast communication may serve as potential therapeutic targets. Additionally, a quantitative presentation of the limitations of the models (e.g the absence of stenosis), potentially including hemodynamic data, could be considered. Regarding the interpretation of differences between human and murine data, it may be helpful to briefly elaborate on how these differences could affect the translational relevance of future preclinical studies.Finally, given the complexity of the study, the main biological messages could be more effectively communicated to both clinical and basic research audiences through a graphical summary (schematic overview).
Overall this is a well-designed, scientifically rigorous, and highly promising study, which I recommend for publication with minor additions.
Author Response
Q. It would be valuable to discuss in more detail the functional validation of the identified macrophage phenotypes and which components of macrophage-fibroblast communication may serve as potential therapeutic targets.
A. We now expand in the Discussion about various components of the fibroblast-macrophage crosstalk that may be worth pursuing in mechanistic studies due to their potential translational relevance to AVF stenosis in humans (page 15, paragraph 3). The lack of functional validation of these cell-to-cell interactions in the present study was also listed as a limitation in the corresponding section of the Discussion (page 16, paragraph 2).
Q. Additionally, a quantitative presentation of the limitations of the models (e.g., the absence of stenosis), potentially including hemodynamic data, could be considered.
A. The new Figure S1 presents the blood flows of contralateral jugular veins and of early, mid, and late AVFs measured with a Transonic probe before tissue harvest. Mid and late AVFs have significantly higher flows than contralateral veins. As suggested by the Reviewer, we acknowledge in the Discussion that one of the main limitations of animal models is the absence of failure (page 16, paragraphs 1 and 2). Arteriovenous fistula maturation failure is a clinical definition based upon the inability of the access to support hemodialysis. This cannot be extrapolated to small animal models. Murine models are useful for mechanistic studies using surrogate parameters of venous outward remodeling such as wall thickness, luminal area, and flow.
Q. Regarding the interpretation of differences between human and murine data, it may be helpful to briefly elaborate on how these differences could affect the translational relevance of future preclinical studies.
A. We agree. We now comment in the Discussion about the potential reasons for the differences between the human and murine data. We also discuss the importance of accounting for or acknowledging these sources of molecular and cellular variation to guide the interpretation of future pre-clinical studies (pages 15-16).
Q. Finally, given the complexity of the study, the main biological messages could be more effectively communicated to both clinical and basic research audiences through a graphical summary (schematic overview).
A. The resubmitted materials include a graphical abstract summarizing the design and main findings of the study.
Reviewer 2 Report
Comments and Suggestions for Authors
Review Report
Title:
Sequential Inflammatory and Matrisome Programs Drive Remodeling of the Mouse Carotid–Jugular Arteriovenous Fistula
Authors:
Filipe F. Stoyell-Conti et al.
General Comments
This manuscript presents an interesting and carefully conducted study on vascular remodeling in a mouse carotid–jugular arteriovenous fistula (AVF) model. The authors explore the temporal progression of inflammatory and extracellular matrix–related programs using both bulk and single-cell RNA sequencing. The addition of human AVF data enhances the translational relevance of the findings.
Overall, I found the study scientifically solid, well written, and relevant to the field. The data are convincing, and the conclusions are generally supported. The paper fits well within the scope of Cells and will likely be of interest to researchers studying vascular remodeling and dialysis access outcomes.
Major Points
- Model limitations and translational relevance
Although the mouse AVF model is valuable, it cannot fully replicate the complexity of human AVFs, particularly in relation to stenosis and hemodynamic factors.
I recommend that the authors discuss these limitations more explicitly and comment on how their mouse data relate to human AVF maturation and failure. - Validation of key findings
Some of the identified signaling pathways and cell–cell interactions (e.g., between macrophages and fibroblasts) are intriguing but remain correlative.
The authors could strengthen the manuscript by providing some experimental validation or by discussing possible follow-up approaches, such as pathway inhibition or knockout models. - Integration of bulk and single-cell data
The integration strategy between bulk and single-cell RNA-seq datasets is mentioned only briefly.
It would be helpful if the authors could describe how the datasets were harmonized and how cross-validation between mouse and human data was achieved.
Minor Points
- Figures are generally clear, but some heatmaps and supplementary figures would benefit from more descriptive legends.
- The human sample set appears relatively small; a note on the limitations of this dataset would add transparency.
- Some terms describing remodeling stages and cell populations could be used more consistently throughout the text.
Author Response
Q. Model limitations and translational relevance: Although the mouse AVF model is valuable, it cannot fully replicate the complexity of human AVFs, particularly in relation to stenosis and hemodynamic factors. I recommend that the authors discuss these limitations more explicitly and comment on how their mouse data relate to human AVF maturation and failure.
A. We agree and thank the Reviewer for stressing this important point. In response to this comment and a similar question by Reviewer 1, we now explicitly discuss the value and limitations of the mouse model to study AVF remodeling. We also advise on how to leverage the availability of both human and mouse single cell data to dissect the mechanisms of maturation and failure (page 16, paragraph 1).
Q. Validation of key findings: Some of the identified signaling pathways and cell–cell interactions (e.g., between macrophages and fibroblasts) are intriguing but remain correlative. The authors could strengthen the manuscript by providing some experimental validation or by discussing possible follow-up approaches, such as pathway inhibition or knockout models.
A. Thank you for this important comment. The overarching goal of this manuscript is to provide the scientific community with a single-cell RNA sequencing database derived from a mechanistically relevant model of postoperative venous remodeling. Our intention was to present these data in the most unbiased manner possible. We hope the Reviewer understands that, given the large number of predicted cell-to-cell interactions and the complexity of the implicated pathways, a comprehensive functional validation of these interactions is beyond the scope of the present study. We acknowledge this limitation and explicitly state it in the Limitations section of the Discussion (page 16, paragraph 2). Furthermore, informed by recently published single-cell associations with AVF maturation failure in humans (Martinez et al., Kidney International, 2025), we expand in the Discussion on specific components of the fibroblast-macrophage crosstalk that may merit investigation in future mechanistic and functional studies (page 15, paragraph 3).
Q. Integration of bulk and single-cell data: The integration strategy between bulk and single-cell RNA-seq datasets is mentioned only briefly. It would be helpful if the authors could describe how the datasets were harmonized and how cross-validation between mouse and human data was achieved.
A. It is now clearly explained in the Materials and Methods that bulk and single-cell RNA-seq datasets were analyzed independently, without data deconvolution of bulk RNA sequences. This allowed an orthogonal validation of the upregulated genes after anastomosis. The single cell analysis showed the postoperative changes in gene expression by cell type, while the bulk sequencing data presents the RNA abundance of specific genes relative to other RNAs in the tissue and the magnitude of the change at the tissue level. Similarly, when comparing the genes upregulated in mice and humans after anastomosis, we included the entire macrophage cluster and myofibroblast/fibroblast cluster from both species. Since all clusters in both species were defined by the same conserved markers and we were not studying specific subpopulations, there was no need for cross-species harmonization. This clarification is now in sections 2.2 and 2.3 of the Materials and Methods.
Q. Figures are generally clear, but some heatmaps and supplementary figures would benefit from more descriptive legends.
A. Thank you for pointing this out. We have carefully revised the figure legends and expanded where necessary.
Q. The human sample set appears relatively small; a note on the limitations of this dataset would add transparency.
A. As requested, we have included this limitation in the corresponding section of the Discussion (page 16, paragraph 2). We agree that the human dataset including 6 pre-access veins, 2 early AVF resections, and 12 transposition AVFs is still small. We hope that future multi-center vascular access studies and clinical trials will incorporate omics as part of the readouts, thereby increasing not only the number of patients but, more importantly, the number of captured cells to enable more robust cluster definition and functional analyses.
Q. Some terms describing remodeling stages and cell populations could be used more consistently throughout the text.
A. We have carefully read the text and harmonized the terminology used for cells and remodeling stages. For example, we used “state” in section 3.1 to match the term used for the main remodeling stages in Figure 1. We also replaced mono/mac with mono/macro throughout the manuscript as well as in Figures 2, 3, and 7.